# Expanding the Library of 1,2,4-Oxadiazole Derivatives: Discovery of New Farnesoid X Receptor (FXR) Antagonists/Pregnane X Receptor (PXR) Agonists

**DOI:** 10.3390/molecules28062840

**Published:** 2023-03-21

**Authors:** Claudia Finamore, Carmen Festa, Bianca Fiorillo, Francesco Saverio Di Leva, Rosalinda Roselli, Silvia Marchianò, Michele Biagioli, Lucio Spinelli, Stefano Fiorucci, Vittorio Limongelli, Angela Zampella, Simona De Marino

**Affiliations:** 1Department of Pharmacy, University of Naples “Federico II”, Via D. Montesano 49, 80131 Naples, Italy; 2Department of Pharmacological Sciences, Icahn School of Medicine at Mount Sinai, 1468 Madison Ave, New York, NY 10029, USA; 3Department of Medicine and Surgery, University of Perugia, Piazza L. Severi, 1-06132 Perugia, Italy; 4Faculty of Biomedical Sciences, Euler Institute, Università della Svizzera italiana (USI), Via G. Buffi 13, CH-6900 Lugano, Switzerland

**Keywords:** farnesoid X receptor antagonist, pregnane X receptor agonist, 1,2,4-oxadiazole, inflammatory disorders

## Abstract

Compounds featuring a 1,2,4-oxadiazole core have been recently identified as a new chemotype of farnesoid X receptor (FXR) antagonists. With the aim to expand this class of compounds and to understand the building blocks necessary to maintain the antagonistic activity, we describe herein the synthesis, the pharmacological evaluation, and the in vitro pharmacokinetic properties of a novel series of 1,2,4-oxadiazole derivatives decorated on the nitrogen of the piperidine ring with different N-alkyl and N-aryl side chains. In vitro pharmacological evaluation showed compounds **5** and **11** as the first examples of nonsteroidal dual FXR/Pregnane X receptor (PXR) modulators. In HepG2 cells, these compounds modulated PXR- and FXR-regulated genes, resulting in interesting leads in the treatment of inflammatory disorders. Moreover, molecular docking studies supported the experimental results, disclosing the ligand binding mode and allowing rationalization of the activities of compounds **5** and **11**.

## 1. Introduction

The nuclear farnesoid X receptor (FXR), mainly expressed in enterohepatic tissues, is a well-known target of bile acids (BAs). FXR regulates the expression of many genes encoding proteins involved in the entire metabolic process [1].

Its activation modulates the synthesis, transport, and reabsorption of bile acids and also plays an essential role in glucose and lipid homeostasis. In addition to its role as a metabolic regulator, FXR shows other effects in non-metabolic areas such as liver regeneration, tumorigenesis, and the protection of atherosclerosis and renal diseases [2,3,4,5].

Consequently, FXR represents a promising target for the treatment of metabolic disorders, showing beneficial effects in different diseases such as cholestasis, nonalcoholic steatohepatitis (NASH), obesity, and liver fibrosis.

During the past decade, many efforts were directed towards the identification of novel FXR modulators. Many FXR agonists, featuring steroidal or nonsteroidal scaffolds, have been identified, and many of them are in preclinical and clinical phases for the treatment of different hepatic and metabolic diseases [6,7,8].

However, FXR modulation shows multifaceted effects on metabolic disorders, possibly due to differences in tissue expression and disease state [9]. Therefore, the evaluation of antagonistic activity also needs to be considered in the drug discovery pipeline.

Recent studies demonstrated that the inhibition of intestinal FXR, mediated by the natural antagonist tauro-β-muricholic acid (TβMCA), prevented obesity-related metabolic dysfunction in mice [10,11,12,13]. Ursodeoxycholic acid (UDCA), a hydrophilic bile acid, is the mainstay in the treatment of cholestatic liver disorders. Although its mechanism of action is poorly understood, it is demonstrated that UDCA exerts FXR antagonistic effects [14,15].

Furthermore, FXR antagonists could be useful for the treatment of obstructive cholestasis since FXR activation increases the expression of the intestinal bile acid-binding protein (I-BABP) gene [16] and inhibits CYP7A1 gene expression, raising cholesterol levels [17].

These findings suggest that FXR antagonists represent an alternative to the treatment of hypercholesterolemia and related metabolic disorders.

In this study, further expanding the library of 1,2,4-oxadiazole derivatives previously identified by our research group [18], we have identified novel FXR antagonist molecules endowed surprisingly with PXR agonistic activity.

The Pregnane X receptor (PXR) is a nuclear receptor, also known as a xenobiotic sensor, expressed in the liver, small intestine, colon, and, to a lesser extent, in the brain. It is involved in many metabolic pathways, including bile acids, lipids, and glucose homeostasis, as well as inflammation [19].

As concerning the therapeutic applications, PXR agonists, such as rifaximin, are currently used in the therapy of inflammatory bowel diseases, such as Crohn’s and ulcerative colitis [20].

To date, the first and only example of a dual PXR/FXR modulator is Theonellasterol G, a marine sterol isolated in 2011 from the sponge *Theonella swinhoei* [21].

Since FXR and PXR are both involved in intestinal inflammation, a dual ligand able to activate PXR and modulate FXR activity holds potential in the treatment of inflammatory disorders.

Heterocyclic compounds, found in the structural core of many natural and synthetic medicinal substances, have gained importance in medicinal chemistry and material science considering their ability to behave as bioisosters of esters, amides, and carbamides [22,23,24,25].

Derivatives presenting in their structures heterocyclic scaffolds possess a wide range of biological activities, such as anticancer, antifungal, anti-inflammatory, anti-bacterial, antiviral, antihypertensive, and antioxidant [26].

Among these, 1,2,4-oxadiazoles represent an interesting class of five-membered heterocyclic compounds that could be decorated at positions C-3 and C-5 with different substituents. The facile synthetic accessibility as well as the structural features make these compounds useful in a broad range of applications, from medicinal chemistry, organic synthesis as intermediates, and metal catalysts to the chemistry of polymers [27,28,29].

In our previous work, making modifications around the 1,2,4-oxadiazole core, we explored the influence of different aromatic substituents at C3 and of different positions of the nitrogen in the heterocyclic moiety at C5. The study resulted in the identification of compound **1**, featuring a 3-(2-naphthyl)-1,2,4-oxadiazole core and a piperidine ring (Figure 1), as the most potent and selective FXR antagonist of the library (IC_50_ 0.58 μM). Notably, compound **1** was also endowed with excellent pharmacokinetic properties [18].

Starting from this molecule, we report here the synthesis and the biological evaluation as dual FXR and PXR modulators of a novel series of derivatives characterized by a 3-(2-naphthyl)-5-(4-piperidyl)-1,2,4-oxadiazole core, in which the nitrogen of the piperidine ring is functionalized with different alkyl or benzyl groups (Figure 1). Computational studies disclosed the binding mode to both FXR and PXR of the most promising ligand of the series, paving the way for the rational design of novel nonsteroidal dual modulators of these nuclear receptors.

## 2. Results and Discussion

### 2.1. Synthesis and Evaluation of Biological Activity

Compound **1** was prepared from 2-naphthonitrile **1a** using the amidoxime route followed by coupling with N-Boc-isonipecotic acid (N-Boc-Inp-OH) as previously described [18] (Figure 1). Using different linear or branched alkyl bromides, compounds **2**–**9** were obtained. The treatment of compound **1** with methyl 3-(bromomethyl)benzoate and methyl 4-(bromomethyl)benzoate allowed to obtain the methyl esters **12** and **13**, respectively, that in turn were reduced with diisobutylaluminium hydride (DIBAL-H), affording the corresponding benzyl alcohols **10** and **11**.

^1^H and ^13^C NMR spectra of compounds **2**–**11** are reported in the Appendix A.

The newly synthesized oxadiazoles were evaluated on FXR in a luciferase reporter assay. As shown in Table 1, compounds **2**, **3**, **7**, and **9** unfortunately proved to be toxic on HepG2 cells at 50 μM, while compounds **5**, **10**, and **11** emerged as FXR antagonists with an efficacy of 66%, 66%, and 72%, respectively. The evaluation of the selectivity towards other nuclear receptors, such as the liver X receptor (LXRα and LXRβ, Appendix A) and PXR (Table 1), revealed that compounds **4**, **5**, **8**, and **11** were also able to activate PXR with an efficacy of 68%, 71%, 50%, and 105%, respectively.

Overall, compounds **5** and **11** represent promising dual modulators in terms of efficacy, emerging as potent PXR agonists and FXR antagonists.

The potency of selected compounds was investigated by a detailed measurement of the concentration-response curves on FXR and PXR, as shown in Table 2.

Liquid chromatography–mass spectrometry (LC-MS) analysis was performed to establish the physicochemical properties of the best active compounds **5** and **11**. They exhibited suitable logD values and showed also good metabolic stabilities in the S9 fraction, with half-life (t_1/2_) of 69 and 204 min and intrinsic clearances (Cl_int_) of 33.7 and 11.3, respectively (Table 2).

To give support to the biological behavior of compounds **5** and **11**, we then tested their effects on HepG2 and CaCo2 cell lines using quantitative polymerase chain reaction (qPCR) analysis. As shown in Figure 2, **5** and **11** reversed the effect of 6-ethyl chenodeoxycholic acid (6-ECDCA), a well-known FXR agonist, on the expression of the canonical FXR target gene bile salt export pump (BSEP) in the HepG2 cell line but showed only a trend on organic solute transporter alpha (OSTα) expression. Only compound **5** showed antagonistic activity on the expression of the mediator of BA transport sodium-dependent uptake transporter (NTCP) (panels **A**–**C**). Additionally, panels **D**–**F** in Figure 2 demonstrated that **5** and **11** reduced changes in the production of proinflammatory factors caused by tumor necrosis factor alpha (TNFα) stimulation in CaCo2, as well as rifaximin, a well-known PXR agonist.

### 2.2. Molecular Docking

In order to investigate the binding mode of the PXR/FXR dual-active compounds **5** and **11**, docking calculations were performed [30,31,32] with the Glide software package (see Appendix A for details).

As regards the docking in PXR, we used two different receptor structures that were purposely chosen considering the bulkiness of the ligands in complex with the receptor. In detail, for compound **5**, we used the ligand binding domain (LBD) X-ray structure of PXR with PDB ID 7AXE [33] that was co-crystallized with a small ligand similar in shape to compound **5**, while considering the bulkier para-hydroxymethyl group present in compound **11**, we docked this ligand in the PXR-LBD X-ray structure with PDB ID 3HVL [34] that was indeed resolved in a complex with a larger compound. This strategy was necessary since docking calculations treat proteins as rigid bodies, and backbone or side chain conformational changes induced by the binding of differently sized ligands cannot be simulated during docking.

Docking calculations show for both compounds **5** and **11** a similar binding mode in PXR, albeit with some differences (Figure 3). In particular, the naphthyl scaffold of each ligand is placed in the hydrophobic pocket between H3, H11, and H12, formed by Leu240, Phe251, Phe281, Asn404, His407, Leu411, Phe420, Ala421, Met425, and Phe429, where it establishes a number of favorable contacts such as π-stacking interactions with Phe429; notably, an additional π-π interaction with Phe240 is formed by **5**. The oxadiazole ring of this compound can also form an H-bond with the side chain of Gln285, while the ligand’s piperidine ring contacts Met246 and Phe288 and forms cation-π interactions with Phe288 and Tyr306. Finally, the alkyl chain points toward the β-strand, where it can interact with lipophilic residues such as Leu209, Val211, Met243, Trp299, and Tyr306. Similar to compound **5**, the oxadiazole ring of **11** is located close to Cys284 and Gln285 while the piperidine ring can interact with Leu209, Val211, and Phe288 and makes a cation-π interaction with Tyr306. Finally, the para-hydroxymethyl group points towards H4 and H11, where it forms an H-bond with the backbone of Ile236.

As regards FXR, the crystal structure of the human FXR-LBD (PDB ID 4OIV) [35] in its antagonist-bound (inactive) state was employed, and docking calculations show a very similar binding mode of compounds **5** and **11**. In the best-scoring and most populated pose, both ligands occupy the binding site defined by H3, H5, H6, H7, and H11 (Figure 4). Here, the ligands’ oxadiazole rings engage both a water-mediated hydrogen bond and a π-π stacking interaction with the side chain of His451 on H11 and an additional water-mediated hydrogen bond with the backbone of Tyr365. Furthermore, the naphthyl ring establishes hydrophobic interactions with residues such as Ala295, Val299, Leu455, and Met456. On the other side, the ligand’s piperidine ring establishes favorable interactions with Ser336, Ile356, Ile361, Ile366, and Tyr373. Finally, the alkyl chain of compound **5** and the benzyl ring of compound **11** form hydrophobic contacts with Leu291, Met294, Ala295, and Leu352. Additionally, the para-hydroxymethyl moiety of compound **11** forms a direct H-bond with His298. Interestingly, the binding modes of compound **5** and compound **11** resemble the ones previously described by us for oxadiazole FXR antagonists [18] and the crystallographic binding pose of N-benzyl-N-(3-tert-butyl-4-hydroxyphenyl)-2,6-dichloro-4-(dimethylamino)benzamide at the FXR-LBD (PDB ID 4OIV) [35]. In particular, the interactions with residues known to play an important role in the binding of FXR antagonists, including the water-mediated H-bonds with His298 and Tyr365 and the hydrophobic pattern involving Leu291, Ala295, Val299, and Leu455, are conserved (Figure 4).

Such interactions are known to stabilize the inactive conformation of the FXR-LBD [18], which shows minor but substantial structural differences if compared with the receptor agonistic form. In detail, in the antagonist-bound FXR-LBD, the C-tail of helix H11 is bent, occupying in part the ligand binding pocket, while H12 unfolds, pointing toward the binding site of the coactivator peptides of the neighboring FXR monomer. Such a conformational change is likely due to the capability of FXR antagonist ligands to disrupt the well-characterized cation-π interaction between His451 on H11 and Trp473 on H12, which is crucial for receptor activation. This, in turn, impedes the recruitment of FXR coactivators, eventually blocking the receptor in a non-activable state.

Overall, our results indicate that in FXR compounds with both linear alkyl chains (**4**, **5**) and bulkier benzyl groups (**10**, **11**) as N-piperidine substituents, the antagonistic profiles are good, suggesting a relative plasticity of the receptor in the region hosting such groups. Here, it might be worth investigating additional functionalization of the piperidine nitrogen (e.g., substituents endowed with hydroxyl, aminic, and other polar functional groups). On the other hand, in PXR, the pocket hosting the N-piperidine substituent is narrow and thus linear; thus, not very long substituents are preferred (**4**, **5** vs. **6**, **7**; **11** vs. **10**), with the para-hydroxymethyl benzyl substituent (**11**) showing the best efficacy. The morphology of this pocket in PXR might represent a limitation in the design of FXR/PXR dual active ligands that should be taken into consideration.

## 3. Materials and Methods

### 3.1. Synthesis

#### 3.1.1. General Information

All chemicals were obtained from commercial vendors (Merk-Sigma, KGaA, Darmstadt, Germany; TCI, Tokyo Chemical Industry Co. Alts Nihongashi-honcho, Chou-Ku, Japan and Fluorochem Limited, Hadfield, UK). Flash chromatography was performed using Biotage Selekt (Uppsala, Sweden) and silica gel columns (Agela 40–60 μm, 60A) Phenomenex, Torrence, CA, USA. Solvents and reagents were used as supplied from commercial sources, with the exception of tetrahydrofuran, which was distilled from calcium hydride immediately prior to use, and methanol, which was dried as previously reported [36].

The reactions were carried out under an argon atmosphere, and the progress of reactions was monitored via thin-layer chromatography (TLC) on Alugram^®^ silica gel G/UV254 plates. The purity of the tested compounds was determined by HPLC analysis. All compounds for biological testing were >95% pure. HPLC was performed using a Waters Model 510 pump equipped with a Waters rheodine injector and differential refractometer, Model 401 (Waters Co., Milford, MA, USA). ESI-MS spectra were performed on a mass spectrometer, the LTQ-XL equipped with an HESI source and coupled with an Ultimate 3000 UHPLC system (Thermo Fisher Scientific, Waltham, MA, USA).

NMR spectra were obtained on Bruker Avance NEO 400 and 700 spectrometers (Bruker Company Billerica, MA, USA) equipped with an RT-DR-BF/1H-5 mm-OZ SmartProbe (^1^H at 400 MHz and ^13^C at 100 MHz; ^1^H at 700 MHz and ^13^C at 175 MHz) and recorded in CD_3_OD (δ_H_ = 3.31 and δ_C_ = 49.0 ppm). All detected signals were in accordance with the proposed structures. *J* are reported in hertz (Hz) and chemical shifts (δ) in ppm, referred to as internal standard for CHD_2_OD. Spin multiplicities are given as s (singlet), br s (broad singlet), d (doublet), t (triplet), or m (multiplet).

#### 3.1.2. Synthetic Procedures

3-(Naphthalen-2-yl)-5-(piperidin-4-yl)-1,2,4-oxadiazole (**1**) was prepared as previously reported [18].

**General synthetic procedures to prepare 1,2,4-oxadiazole derivatives 2–9 and 12–13**. N,N-diisopropylethylamine (DIPEA, 94 μL, 0.54 mmol) and alkyl or benzyl bromide (1.5 mol eq.) were added to compound **1** (50 mg, 0.180 mmol) dissolved in acetonitrile dry (5 mL) and stirred at 60 °C over night. After completion of the reaction (monitored by TLC), the resulting solution was then concentrated under vacuum, diluted with water, and extracted with CH_2_Cl_2_. The organic fraction was dried over NaSO_4_, and the solvent was removed under reduced pressure to yield the crude products (50–100% yields).

5-(1-Ethylpiperidin-4-yl)-3-(naphthalen-2-yl)-1,2,4-oxadiazole (**2**). The mixture was purified by HPLC on a Synergi Fusion-RP 80 (4 μm; 4.6 mm i.d. × 250 mm) with MeOH/H_2_O (88:12) as eluent (flow rate 1 mL/min, t_R_ = 6.7 min); ^1^H NMR (400 MHz, CD_3_OD): δ_H_ 8.60 (1H, s), 8.09 (1H, dd, *J* = 8.5, 1.5 Hz), 7.98 (1H, ovl), 7.97 (1H, ovl), 7.93 (1H, br d, *J* = 7.6 Hz), 7.58 (2H, m), 3.16 (1H, m), 2.99 (2H, m), 2.52 (2H, q, *J* = 7.2 Hz), 2.27 (2H, br t, *J* = 11.0 Hz), 2.19 (2H, m), 2.02 (2H, m), 1.16 (3H, t, *J* = 7.2 Hz, CH_3_); ^13^C NMR (100 MHz, CD_3_OD): δ_C_ 183.6, 169.4, 136.2, 134.5, 129.9 (2C), 128.8 (3C), 127.9, 125.5, 124.6, 53.7 (3C), 35.3, 30.1 (2C), 12.5; ESIMS *m/z* 308.4 [M + H]^+^.

3-(Naphthalen-2-yl)-5-(1-propylpiperidin-4-yl)-1,2,4-oxadiazole (**3**). The mixture was purified by HPLC on a Synergi Fusion-RP 80 (4 μm; 4.6 mm i.d. × 250 mm) with MeOH/H_2_O (88:12) as eluent (flow rate 1 mL/min, t_R_ = 8.2 min); ^1^H NMR (700 MHz, CD_3_OD): δ_H_ 8.60 (1H, s), 8.09 (1H, dd, *J* = 8.5, 1.5 Hz), 7.99 (2H, ovl), 7.93 (1H, br d, *J* = 7.6 Hz), 7.58 (2H, m), 3.15 (1H, m), 3.06 (2H, m), 2.41 (2H, m), 2.28 (2H, br t, *J* = 11.0 Hz), 2.20 (2H, m), 2.04 (2H, m), 1.59 (2H, m), 0.95 (3H, t, *J* = 7.4 Hz, CH_3_); ^13^C NMR (175 MHz, CD_3_OD): δ_C_ 183.6, 169.5, 136.1, 134.5, 129.9, 129.8, 128.9, 128.8, 128.7, 127.9, 125.5, 124.6, 61.9, 53.7 (2C), 35.3, 30.1 (2C), 20.6, 12.2; ESIMS *m/z* 322.2 [M + H]^+^.

5-(1-Butylpiperidin-4-yl)-3-(naphthalen-2-yl)-1,2,4-oxadiazole (**4**). The mixture was purified by HPLC on a Synergi Fusion-RP 80 (4 μm; 4.6 mm i.d. × 250 mm) with MeOH/H_2_O (88:12) as eluent (flow rate 1 mL/min, t_R_ = 11.7 min); ^1^H NMR (400 MHz, CD_3_OD): δ_H_ 8.60 (1H, s), 8.09 (1H, dd, *J* = 8.5, 1.5 Hz), 7.98 (1H, ovl), 7.97 (1H, ovl), 7.93 (1H, br d, *J* = 7.6 Hz), 7.58 (2H, m), 3.17 (1H, m), 3.07 (2H, m), 2.45 (2H, m), 2.30 (2H, m), 2.23 (2H, m), 2.05 (2H, m), 1.56 (2H, m), 1.39 (2H, m), 0.97 (3H, t, *J* = 7.2 Hz, CH_3_); ^13^C NMR (100 MHz, CD_3_OD): δ_C_ 183.7, 169.8, 136.1, 134.6, 129.9, 129.8, 128.9, 128.8, 128.7, 127.9, 125.5, 124.6, 59.8, 53.7 (2C), 35.3, 30.2 (2C), 29.7, 21.9, 14.4; ESIMS *m/z* 336.2 [M + H]^+^.

3-(Naphthalen-2-yl)-5-(1-pentylpiperidin-4-yl)-1,2,4-oxadiazole (**5**). The mixture was purified by HPLC on a Synergi Fusion-RP (4 μm; 4.6 mm i.d. × 250 mm) with MeOH/H_2_O (88:12) as eluent (flow rate 1 mL/min, t_R_ = 14.6 min); ^1^H NMR (400 MHz, CD_3_OD): δ_H_ 8.60 (1H, s), 8.09 (1H, dd, *J* = 8.5, 1.3 Hz), 8.00 (1H, ovl), 7.99 (1H, ovl), 7.93 (1H, d, *J* = 7.2 Hz), 7.58 (2H, ovl), 3.14 (1H, m), 3.03 (2H, m), 2.40 (2H, m), 2.21 (4H, m), 2.03 (2H, m), 1.57 (2H, m), 1.37 (2H, m), 1.33 (2H, m), 0.93 (3H, t, *J* = 7.1 Hz, CH_3_); ^13^C NMR (100 MHz, CD_3_OD) δ_C_ 183.5, 169.5, 136.0, 134.4, 129.9, 129.8, 128.9, 128.8, 128.7, 128.0, 125.4, 124.6, 59.0, 53.5 (2C), 35.2,31.0, 30.0 (2C), 27.0, 23.7, 14.4; ESIMS *m*/*z* 350.2 [M + H]^+^.

5-(1-Hexylpiperidin-4-yl)-3-(naphthalen-2-yl)-1,2,4-oxadiazole (**6**). The mixture was purified by HPLC on a Synergi Fusion-RP 80 (4 μm; 4.6 mm i.d. × 250 mm) with MeOH/H_2_O (88:12) as eluent (flow rate 1 mL/min, t_R_ = 14.9 min); ^1^H NMR (400 MHz, CD_3_OD): δ_H_ 8.60 (1H, s), 8.09 (1H, dd, *J* = 8.6, 1.4 Hz), 8.00 (1H, ovl), 7.99 (1H, ovl), 7.93 (1H, d, *J* = 7.2 Hz), 7.58 (2H, ovl), 3.16 (1H, m), 3.05 (2H, m), 2.43 (2H, m), 2.24 (4H, m), 2.05 (2H, m), 1.57 (2H, m), 1.35 (6H, m), 0.93 (3H, t, *J* = 7.1 Hz, CH_3_); ^13^C NMR (100 MHz, CD_3_OD): δ_C_ 183.6, 169.5, 136.1, 134.5, 129.9, 129.8, 128.9, 128.8, 128.7, 127.9, 125.5, 124.6, 60.0, 53.7 (2C), 35.5, 32.9, 30.3, 30.1 (2C), 28.6, 27.4, 14.3; ESIMS *m/z* 364.2 [M + H]^+^.

5-(1-Heptylpiperidin-4-yl)-3-(naphthalen-2-yl)-1,2,4-oxadiazole (**7**). The mixture was purified by HPLC on a Synergi Fusion-RP 80 (4 μm; 4.6 mm i.d. × 250 mm) with MeOH/H_2_O (88:12) as eluent (flow rate 1 mL/min, t_R_ = 15.4 min); ^1^H NMR (700 MHz, CD_3_OD): δ_H_ 8.60 (1H, s), 8.09 (1H, dd, *J* = 8.7, 1.3 Hz), 8.00 (1H, ovl), 7.99 (1H, ovl), 7.93 (1H, d, *J* = 7.2 Hz), 7.58 (2H, ovl), 3.16 (1H, m), 3.06 (2H, m), 2.44 (2H, m), 2.28 (2H, m), 2.22 (2H, m), 2.05 (2H, m), 1.57 (2H, m), 1.35 (8H, ovl), 0.92 (3H, t, *J* = 7.1 Hz, CH_3_); ^13^C NMR (175 MHz, CD_3_OD): δ_C_ 183.6, 169.5, 136.1, 134.5, 129.9, 129.8, 128.9, 128.8, 128.7, 127.9, 125.5, 124.6, 60.0, 53.7 (2C), 35.4, 32.9, 30.3, 30.1 (2C), 28.7, 27.6, 23.7, 14.5; ESIMS *m/z* 378.2 [M + H]^+^.

5-(1-Isopropylpiperidin-4-yl)-3-(naphthalen-2-yl)-1,2,4-oxadiazole (**8**). The mixture was purified by HPLC on a Synergi Fusion-RP 80 (4 μm; 4.6 mm i.d. × 250 mm) with MeOH/H_2_O (95:5) as eluent (flow rate 1 mL/min, t_R_ = 6.2 min); ^1^H NMR (700 MHz, CD_3_OD): δ_H_ 8.60 (1H, s), 8.09 (1H, dd, *J* = 8.6, 1.3 Hz), 8.00 (1H, ovl), 7.99 (1H, ovl), 7.94 (1H, d, *J* = 7.5 Hz), 7.58 (2H, ovl), 3.14 (1H, m), 3.04 (2H, m), 2.81 (1H, septet), 2.46 (2H, m), 2.23 (2H, m), 2.02 (2H, m), 1.13 (6H, d, *J* = 6.6 Hz, CH_3_); ^13^C NMR (175 MHz, CD_3_OD): δ_C_ 183.6, 169.5, 136.1, 134.5, 129.9, 129.8, 128.9, 128.8, 128.7, 127.9, 125.5, 124.6, 56.2, 35.6, 48.8 (2C), 30.4 (2C), 18.3 (2C); ESIMS *m/z* 322.2 [M + H]^+^.

5-(1-(Sec-butyl)piperidin-4-yl)-3-(naphthalen-2-yl)-1,2,4-oxadiazole (**9**). The mixture was purified by HPLC on a Synergi Fusion-RP 80 (4 μm; 4.6 mm i.d. × 250 mm) with MeOH/H_2_O (88:12) as eluent (flow rate 1 mL/min, t_R_ = 13.4 min); ^1^H NMR (400 MHz, CD_3_OD): δ_H_ 8.60 (1H, s), 8.09 (1H, dd, *J* = 8.6, 1.3 Hz), 8.00 (1H, ovl), 7.99 (1H, ovl), 7.94 (1H, d, *J* = 7.5 Hz), 7.59 (2H, ovl), 3.13 (1H, m), 2.98 (2H, m), 2.58 (2H, m), 2.49 (1H, m), 2.22 (2H, m), 2.02 (2H, m), 1.69 (1H, m), 1.35 (1H, m), 1.07 (3H, d, *J* = 6.6 Hz, CH_3_), 0.96 (3H, t, *J* = 7.3 Hz, CH_3_); ^13^C NMR (100 MHz, CD_3_OD): δ_C_ 183.6, 169.5, 136.1, 134.5, 129.9, 129.8, 128.9, 128.8, 128.7, 127.9, 125.5, 124.6, 59.5, 54.8 (2C), 35.6, 33.6 (2C), 25.4, 13.6, 12.0; ESIMS *m/z* 336.2 [M + H]^+^.

5-[1-(3-(Methoxycarbonyl)benzyl)piperidin-4-yl]-3-(naphthalen-2-yl)-1,2,4-oxadiazole (**12**). ^1^H NMR (700 MHz, CD_3_OD): δ_H_ 8.60 (1H, s), 8.09 (1H, dd, *J* = 8.5, 1.5 Hz), 8.05 (1H, s), 7.98 (2H, ovl), 7.94 (2H, t, *J* = 7.6 Hz), 7.62 (1H, d, *J* = 7.6 Hz), 7.58 (2H, ovl), 7.47 (1H, t, *J* = 7.6 Hz), 3.91 (3H, s, OCH_3_), 3.63 (2H, s), 3.19 (1H, m), 3.07 (2H, m), 2.43 (2H, m), 2.23 (2H, m), 2.06 (2H, m); ^13^C NMR (175 MHz, CD_3_OD): δ_C_ 183.4, 169.5, 168.5, 136.1, 135.6, 134.8, 134.7, 130.9, 129.9, 129.8, 129.7, 129.6, 129.3, 128.9, 128.8, 128.7, 128.0, 125.5, 124.6, 63.3, 53.4 (2C), 52.6, 35.1, 30.0 (2C), ESIMS *m/z* 428.2 [M + H]^+^.

5-[1-(4-(Methoxycarbonyl)benzyl)piperidin-4-yl]-3-(naphthalen-2-yl)-1,2,4-oxadiazole (**13**). ^1^H NMR (700 MHz, CD_3_OD): δ_H_ 8.60 (1H, s), 8.09 (1H, dd, *J* = 8.5, 1.5 Hz), 8.02 (2H, d, *J* = 8.3 Hz), 8.01 (1H, ovl), 7.99 (1H, ovl), 7.94 (1H, br d, *J* = 7.6 Hz), 7.58 (2H, m), 7.51 (2H, d, *J* = 8.3 Hz), 3.91 (3H, s, OCH_3_), 3.65 (2H, s), 3.16 (1H, m), 2.99 (2H, m), 2.31 (2H, br t, *J* = 10.9 Hz), 2.19 (2H, m), 2.05 (2H, m); ^13^C NMR (175 MHz, CD_3_OD): δ_C_ 183.6, 169.5, 168.5, 144.8, 136.1, 134.5, 130.6 (2C), 130.5 (2C), 130.4, 129.9, 129.8, 128.9, 128.8, 128.7, 128.0, 125.5, 124.6, 63.6, 53.7 (2C), 52.6, 35.4, 30.4 (2C), ESIMS *m/z* 428.2 [M + H]^+^.

**General synthetic procedures to prepare 1,2,4-oxadiazole derivatives 10–11**.

Compounds **12** and **13** (61 mg, 0.144 mmol) were subjected to reduction with DIBAL-H (1.0 M in toluene, 290 μL, 0.288 mmol) added dropwise in a solution with tetrahydrofuran at 0 °C. The reaction was stirred at room temperature for 12–24 h. The reaction was quenched by slow addition of methanol and of a solution of saturated Rochelle salt, stirred for 1 h. The mixture was partitioned three times, and the combined organic extracts dried over Na_2_SO_4_.

5-[1-(3-(Hydroxymethyl)benzyl)piperidin-4-yl]-3-(naphthalen-2-yl)-1,2,4-oxadiazole (**10**). The mixture was purified by HPLC on a Synergi Fusion-RP 80 (4 μm; 4.6 mm i.d. × 250 mm) with MeOH/H_2_O (78:22) as eluent (flow rate 1 mL/min, t_R_ = 19.2 min); ^1^H NMR (400 MHz, CD_3_OD): δ_H_ 8.60 (1H, s), 8.09 (1H, d, *J* = 8.5 Hz), 7.98 (2H, ovl), 7.93 (1H, br d, *J* = 7.6 Hz), 7.58 (2H, m), 7.38 (1H, br s), 7.32 (1H, ovl), 7.29 (2H, ovl), 4.62 (2H, s), 3.60 (2H, s), 3.14 (1H, m), 3.00 (2H, m), 2.28 (2H, br t, *J* = 11.0 Hz), 2.19 (2H, m), 2.04 (2H, m); ^13^C NMR (100 MHz, CD_3_OD): δ_C_ 183.4, 169.5, 142.0, 136.1, 135.5, 134.8 (2C), 131.4, 130.9, 129.9, 129.8, 129.3, 128.9, 128.8, 128.7, 128.0, 125.2, 124.6, 64.0, 63.3, 53.4 (2C), 35.1, 29.9 (2C); ESIMS *m/z* 400.2 [M + H]^+^.

5-[1-(4-(Hydroxymethyl)benzyl)piperidin-4-yl]-3-(naphthalen-2-yl)-1,2,4-oxadiazole (**11**). The mixture was purified by HPLC on a Synergi Fusion-RP 80 (4 μm; 4.6 mm i.d. × 250 mm) with MeOH/H_2_O (80:20) as eluent (flow rate 1 mL/min, t_R_ = 14.3 min); ^1^H NMR (700 MHz, CD_3_OD): δ_H_ 8.60 (1H, s), 8.09 (1H, dd, *J* = 8.5, 1.5 Hz), 7.98 (1H, ovl), 7.97 (1H, ovl), 7.93 (1H, br d, *J* = 7.6 Hz), 7.58 (2H, m), 7.35 (4H, ovl), 4.61 (2H, s), 3.59 (2H, s), 3.13 (1H, m), 2.99 (2H, m), 2.27 (2H, br t, *J* = 11.0 Hz), 2.19 (2H, m), 2.02 (2H, m); ^13^C NMR (175 MHz, CD_3_OD): δ_C_ 183.7, 169.5, 142.0, 137.4, 136.1, 134.5, 130.8 (2C), 129.9, 129.8, 128.9, 128.8, 128.7, 128.0 (2C), 127.9, 125.5, 124.6, 64.9, 63.8, 53.5 (2C), 35.4, 30.2 (2C); ESIMS *m/z* 400.2 [M + H]^+^.

### 3.2. Biological Assays

#### 3.2.1. Luciferase Reporter Gene Assay

The antagonistic activity on FXR was investigated using HepG2 cells transfected with 100 ng of pSG5-FXR and pSG5-RXR, 200 ng of the reporter vector p(hsp27)-TK-LUC containing the FXR response element, and 100 ng of the pGL4.70 vector encoding the human Renilla gene used as a transfection control. Cells were stimulated with the agonist 6-ECDCA (10 µM) alone or in combination with compounds **2**–**11** after 24 h. In another experimental setting, HepG2 cells were transfected using 100 ng of pSG5-PXR, 100 ng of pSG5-RXR, and 100 ng of pGL4.70, along with the reporter vector pGL3(henance)PXRE. At 24 h post-transfection, cells were stimulated with rifaximin 10 μM (as a positive control) or compounds **2**–**11** (10 μM). The specificity of compounds **5** and **11** versus FXR and PXR was tested using HepG2 cells transiently transfected with 200 ng of the reporter vector p(UAS)5XTKLuc, 100 ng of a vector containing the ligand binding domain of LXRα or LXRβand 100 ng of pGL4.70, a vector encoding the human Renilla gene. At 24 h post-transfection, cells were stimulated for 18 h with GW3965 (5 μM), compounds **5** and **11** (10 μM), and the combination of GW3965 plus **5** or **11** (50 μM). Cellular lysate was assayed for luciferase and Renilla activities after 18 h using the Dual-Luciferase Reporter assay system (E1980, Promega, Madison, WI, USA). The activity of luciferase was measured using the Glomax 20/20 luminometer (Promega, Madison, WI, USA) and normalized with Renilla activities.

#### 3.2.2. Cell Culture

HepG2 and CaCo2 cell lines, obtained from ATCC, were routinely cultured in Eagle’s MEM and high-glucose DMEM, respectively, supplemented with 10% FBS, 1% glutamine, and penicillin/streptomycin. For in vitro experiments, HepG2 (2 × 106) cells were stimulated with 6-ECDCA at a concentration of 10 µM alone or in combination with compound **5** or **11**, and CaCo2 cells were exposed to TNFα (100 ng/mL) alone or in combination with rifaximin, compound **5**, or **11**.

#### 3.2.3. Reverse Transcription of mRNA and Real-Time PCR

Total RNA was isolated from the HepG2 and CaCo2 cell lines using TRI reagent (Zymo Research) and Direct-zol™ RNA MicroPrep w/Zymo-Spin™ IIC Columns (Zymo Research, Irvine, CA, USA). After purification from genomic DNA using DNase I (Thermo Fisher Scientific, Waltham, MA, USA), 2 μg of RNA from each sample was reverse transcribed using Kit FastGene Scriptase Basic (Nippon Genetics, Mariaweilerstraße, Düren, Germany) in 20-μL of reaction volume, and 50 ng of cDNA was amplified in a 20-μL solution containing 200 nM of each primer and 10 μL of PowerUp™ SYBR™ Green Master Mix (Thermo Fisher Scientific, Waltham, MA, USA) using a Quantstudio 3 system (Applied Biosystems, Foster City, CA, USA). The relative mRNA expression was calculated according to the ΔCt method. The primer used were as follows (forward and reverse): hGapdh (for CAGCCTCAAGATCATCAGCA; rev GGTCATGAGTCCTTCCACGA), hBsep (for TCCAGGAAAAGCATGTGTGA; rev CATTTCGCTCTCGATGTTCA), hOstα (for TGTTGGGCCCTTTCCAATAC; rev GGCTCCCATGTTCTGCTCAC), hNtcp (for TCAAGACTCCCAAGGATAAAACA; rev ATGTGGAAATGCTGGAGAAA), hPxr (for GCAAGGGCTTTTTCAGGAG; rev GGGTCTTCCGGGTGATCT), hIl-8 (for GCAGCTCTGTGTGAAGGTGCA; rev GTTGGCGCAGTGTGGTCCACT), hIl-1β (for GTGGCAATGAGGATGACTTG; rev GGAGATTCGTAGCTGGATGC).

### 3.3. Physicochemical Properties and Pharmacokinetic Characterization

#### 3.3.1. S9 Stability of Compounds **5** and **11**

Human liver S9 fraction (Sigma-Aldrich, St. Louis, MO, USA) was used. All incubations were performed in duplicate in a thermoblock (Dlab dry bath HB 120-S) at 37 °C. The incubation mixtures contained 1 μM compound with 0.1% DMSO used as a vehicle, human liver S9 fraction (0.3 mg of protein per mL), 5 mM taurine, 5 mM glycine, 5 mM glucose 6-phosphate, 5 mM GSH, 1 mM NADPH, 1 mM ATP, 0.4 mM CoA, 0.4 mM UDPGA, 0.1 mM PAPS, and 0.4 U·mL^−1^ glucose 6-phosphate dehydrogenase. All the cofactors were dissolved in a 50 mM potassium phosphate buffer (pH 7.4), containing 5 mM MgCl_2_, for a final volume of 0.25 mL. Aliquots of 25 µL were removed at 0, 5, 15, 30, 45, 60, and 90 min after S9 addition, and the reaction was stopped by adding 100 μL of ice-cold MeOH. The samples were centrifuged for 10 min at 10,000 rpm, and the supernatants were transferred into vials for LC-MS/MS analysis. The slope of the linear regression of the curve obtained by reporting the natural logarithm of compound area versus incubation time (−k) was used in the conversion to in vitro t_1/2_ values, calculated as t_1/2_ = −ln(2)/k. In vitro intrinsic clearance (Clint expressed as μL/min/mg) was obtained by first calculating V using the formula: V = volume of reaction (μL)/protein of liver microsomes (mg); intrinsic clearance was then calculated as: Clint = (V × ln2)/t_1/2_.

#### 3.3.2. LogD Measurement

40 μL of a 10 mM solution in DMSO of each compound was added to a 2 mL test tube containing 980 μL of PBS pH 7.4 and 980 μL of 1-Octanol (in duplicate). The tubes were shaken for 2 h at room temperature and subsequently kept still to allow for an optimal phase separation. The two phases were then separated, and 10 μL of each sample was diluted in 490 μL of MeOH for LC-MS/MS analysis. LogD was calculated as Log10 of the ratio of mass signal octanol/PBS. LC-MS/MS analysis was conducted using a Kinetex 2.6 µm C18 100 Å LC column 100 × 2.1 mm by means of a linear gradient of the aqueous phase (H_2_O 0.1% FA) and organic phase (ACN 0.1% FA).

### 3.4. Molecular Docking

#### 3.4.1. Receptor Preparation

**PXR**. The crystal structures of the homo sapiens PXR-LBD in complex with agonists, oxadiazon and [2-(3,5-di-tert-butyl-4-hydroxy-phenyl)-1-(diethoxy-phosphoryl)-vinyl]-phosphonic acid diethyl ester (SRL) (PDB IDs 7AXE and 3HVL, respectively) [33,34], were retrieved from the Protein Data Bank website.

**FXR**. The crystal structure of the homo sapiens FXR-LBD in complex with the antagonist N-benzyl-N-(3-(tert-butyl)-4-hydroxyphenyl)-2,6-dichloro-4-(dimethylamino) benzamide (NDB) (PDB ID 4OIV) [34] was retrieved from the Protein Data Bank website as the only tridimensional structure of the inactive form of the FXR-LDB receptor.

The receptor structures were prepared using the Protein Preparation Wizard [37] as implemented in Maestro. Bond orders were assigned, and missing hydrogens were added. A prediction of the receptor side chain ionization and tautomeric states was performed using Epik [38]. However, both the tautomeric states of His298 in FXR were considered in the docking calculations. Then, an optimization of the hydrogen-bonding network was carried out, and the positions of the hydrogen atoms were minimized using OPLS2005 force field [39]. Finally, water molecules beyond 3 Å from the co-crystallized ligand—NBD—in FXR were removed, whereas the others were explicitly considered in the docking calculations as it is reported in the literature that they are involved in the binding mode of other antagonists. As regards PXR, all water molecules were removed.

#### 3.4.2. Ligand Preparation

Ligand 3D structures were built using the Graphical User Interface (GUI) of Maestro ver. 13.3, [40] considering all the four conformers of the piperidine ring. All conformers were then refined using LigPrep [41] as implemented in Maestro. In particular, the protonation states of **5** and **11** at pH 7.4 in water were calculated using the Epik module [42,43]. Finally, the ligands were then minimized using the OPLS2005 force field through 2500 iteration steps of the Polak-Ribiere Conjugate Gradient (PRCG) algorithm [44].

#### 3.4.3. Docking Calculations

Docking calculations in PXR were performed using the induced fit docking (IFD) [45] protocol of the Schrödinger software release 2022-3 [46], which uses the OPLS2005 force field and a combination of Glide and Prime [47,48] to allow optimization of the conformations of the residue side chains in the receptor structure during ligand docking. However, we note that major conformational rearrangements are not sampled using such a protocol; in such a case, the use of multiple receptor structures is advised to explore a larger conformational space. As regards docking calculations in FXR, a grid box of 25 Å × 25 Å × 25 Å centered on the ligand binding cavity was created, and the docking calculation was carried out using the Glide software package [49] with the Standard Precision (SP) algorithm of the GlideScore function. The OPLS2005 force field was employed.

Binding poses were generated and docking conformations of compounds **5** and **11** were then clustered based on their atomic RMSD average. Four and seven clusters were obtained for PXR and FXR, respectively, and, among them, only the conformation included in the most populated cluster with both the Glide Emodel and GlideScore lowest-energy values was considered (Figure 3 and Figure 4).

All figures were rendered by UCSF Chimera [50].

## 4. Conclusions

In conclusion, in this paper, a new series of 1,2,4-oxadiazole compounds has been synthesized and evaluated for their in vitro activities on FXR and PXR. This study resulted in the identification of compounds **5** and **11**, as dual FXR antagonists and PXR agonists.

These two most active derivatives were characterized by the presence, on the nitrogen of the piperidine ring, of either a linear five-carbon chain or a benzyl group bearing a hydroxymethyl in para position. The ability of compounds **5** and **11** to revert the expression of both FXR target genes, involved in bile acid excretion, and inflammatory genes makes these compounds promising leads for the treatment of inflammatory disorders. There have been few examples of dual FXR/PXR modulators to date. Therefore, the discovery of a new chemotype endowed with PXR agonist and FXR antagonist activity represents an attractive opportunity in the field of chronic inflammatory and metabolic diseases.

## Data Availability

Data are contained within the article or Appendix A.

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
