# Peer review of "Expanding the Library of 1,2,4-Oxadiazole Derivatives: Discovery of New Farnesoid X Receptor (FXR) Antagonists/Pregnane X Receptor (PXR) Agonists"

_molecules, 2023, doi:10.3390/molecules28062840_

Round 1

Reviewer 1 Report

There is no literature review for "1,2,4-oxadiazole". Please provide more literature about the significance of oxadiazoles.

Please compare the biological activity of oxadiazole with other five-membered heterocycles. https://www.ingentaconnect.com/content/ben/mrmc/2021/00000021/00000005/art00003

I don't think it is suitable to start the abstract with "In a recent paper".

If possible please provide some products with functional groups such as NO2, Cl and OH.

Author Response

Comments:

  • There is no literature review for "1,2,4-oxadiazole". Please provide more literature about the significance of oxadiazoles.

Please compare the biological activity of oxadiazole with other five-membered heterocycles. https://www.ingentaconnect.com/content/ben/mrmc/2021/00000021/00000005/art00003

Response: Thank you for pointing out this observation in the manuscript. The revised version includes new references (22-29) and a discussion on the use of heterocycles, particularly 1,2,4-oxadiazoles, in medicinal and organic chemistry (line 84-95).

  • I don't think it is suitable to start the abstract with "In a recent paper".

Response: Thank you for the comment. The first sentence of abstract has been carefully revised (line 19-21).

  • If possible, please provide some products with functional groups such as NO2, Cl and OH.

Response: Thank you for your valuable suggestion. We will take in account the possibility to introduce new functionalities, as suggested, in a next library, with the aim to introduce new pharmaphoric points and groups able to establish key interactions in the active pocket of target.

Reviewer 2 Report

Expanding the library of 1,2,4-oxadiazole derivatives: discovery 2 of new FXR antagonists/PXR agonists This manuscript focuses on the design and synthesis of a new series of 1,2,4-oxadiazole compounds. These compounds have been evaluated for their in vitro activities on FXR and PXR. In this study the authors have identified a new chemotype of Farnesoid X receptor (FXR) antagonists featuring a 1,2,4-oxadiaxole core substituted at C3 with a naphthyl and at C5 with a piperidine ring. With the aim to expand this new class of FXR antagonists and to understand the building blocks necessary to maintain the antagonistic activity. The discovery of a new chemotype endowed with PXR agonist and FXR antagonist activity represents an attractive opportunity in the field of chronic inflammatory and metabolic diseases. The purity, composition and structure of the compounds were determined using a variety of techniques, NMR, and ESI-MS. The preparations and characterisations (H-, and CNMR) are described. This is acceptable for organic compounds. The Chapter Results and Discussion is of good scientific quality and the rich and instructive graphic realizes the understanding of the obtained results and of their significance. The experimental data is described appropriately and the manuscript needs no language and grammar corrections. The manuscript is written straight forward. The study is a meaningful suppliment to the series of publications regarding the heterocyclic compounds (with P, S, N atoms) related to natural products: synthesis, structural analysis and investigation of their biological activity, that have been extensively studied because their important properties and applications, especially in biological activities, such as, anti-microbial, anti-proliferative (prostate cancer cells), anticancer , anti-influenza and with antioxidant activity. In Introduction the autors did not reflect any other field of another heterocyclic with the important applications as chiral ligands for metal catalyst or receptors especially in biological activities. You have to describe in the introduction more generally to the use of heterocyclic derivatives with biological activity. Therefore, you should cite more special topics. It is of interest for synthetic chemists. It is recommended to the authors to cite these papers in their introduction a wider base: The medicinal chemistry of gold complexes as anticancer drugs, Ott, I. On, Coord. Chem. Rev. 2009, 253, 1670–1681; N-Heterocyclic Carbenes (NHC) Derived from Imidazo[1,5-a]pyridines Related to Natural Products: Synthesis, Structure and Potential Biological Activity of Some Corresponding Gold(I) and Silver(I) Complexes, Monica Mihorianu, M. Heiko Franz, Peter G. Jones, Matthias Freytag, Gerhard Kelter, Heinz-Herbert Fiebig, Matthias Tamm and Ion Neda , Appl. Organometal. Chem. 2016, 30, 581-589 Trifluoromethylpyridine-Substituted N-Heterocyclic Carbenes (NHC) related to Natural Products: Synthesis, Structure and Potential Antitumor Activity of some Corresponding Gold(I), Rhodium(I) and Iridium(I) Complexes, Elena Maftei, Catalin V. Maftei, Peter G. Jones, Matthias Freytag, M. Heiko Franz, Gerhard Kelter, Heinz-Herbert Fiebig, Matthias Tamm and Ion Neda, Helv. Chim. Acta, 2016, 99, 469-481 Novel 1,2,4-oxadiazoles and trifluoromethylpyridines related to natural products: Synthesis, structural analysis and investigation of their antitumor activity, Maftei, C.V., Fodor, E., Jones, P.G., Daniliuc, C.G., Franz, M.H., Kelter, G., Fiebig, H.-H., Tamm, M., Tetrahedron, 2016,72,1185-1199 In conclusion of my review, I recommend this manuscript for publication with minor revisions

Author Response

Response: Your comment is greatly appreciated. We have added the suggested and other relevant references in the field of heterocycles, privileged scaffolds used in broad range of applications both in medicinal and organic chemistry.

Round 2

Reviewer 1 Report

I have no more comments and the article can be accepted. 

Author Response

Thanks